# Minigrid & Miniworld: Modular & Customizable Reinforcement Learning Environments for Goal-Oriented Tasks

**Maxime Chevalier-Boisvert**
Mila - Québec AI Institute
maximechevalierb@gmail.com

**Bolun Dai**
New York University
& Farama Foundation
bolundai@nyu.edu

**Mark Towers**
University of Southampton
& Farama Foundation
mt5g17@soton.ac.uk

**Rodrigo de Lazcano**
Farama Foundation
rperezvicente@farama.org

**Lucas Willems**
Miple
lucas.willems@miple.co

**Salem Lahlou**
Mila - Québec AI Institute
lahlosal@mila.quebec

**Suman Pal**
Telekinesis
suman7495@gmail.com

**Pablo Samuel Castro**
Google DeepMind
psc@google.com

**Jordan Terry**
Farama Foundation
& Swarm Labs
jkterry@umd.edu

## Abstract

We present the Minigrid and Miniworld libraries which provide a suite of goal-oriented 2D and 3D environments. The libraries were explicitly created with a minimalistic design paradigm to allow users to rapidly develop new environments for a wide range of research-specific needs. As a result, both have received widescale adoption by the RL community, facilitating research in a wide range of areas. In this paper, we outline the design philosophy, environment details, and their world generation API. We also showcase the additional capabilities brought by the unified API between Minigrid and Miniworld through case studies on transfer learning (for both RL agents and humans) between the different observation spaces. The source code of Minigrid and Miniworld can be found at https://github.com/Farama-Foundation/Minigrid and https://github.com/Farama-Foundation/Miniworld along with their documentation at https://minigrid.farama.org/ and https://miniworld.farama.org/.

## 1 Introduction

The capabilities of reinforcement learning (RL) agents have grown rapidly in recent years, in part thanks to the development of deep reinforcement learning (DRL) algorithms [32, 33]. This has been supported by suites of simulation environments such as OpenAI Gym [5] (now `gymnasium`) and `dm_control` [38] that provide common benchmarks for comparing algorithms. These libraries focus on providing environments where the agent learns to control itself or understand complex visual observations (e.g., swinging up a pendulum or playing video games) rather than logical reasoning or instruction following.

37th Conference on Neural Information Processing Systems (NeurIPS 2023) Track on Datasets and Benchmarks.

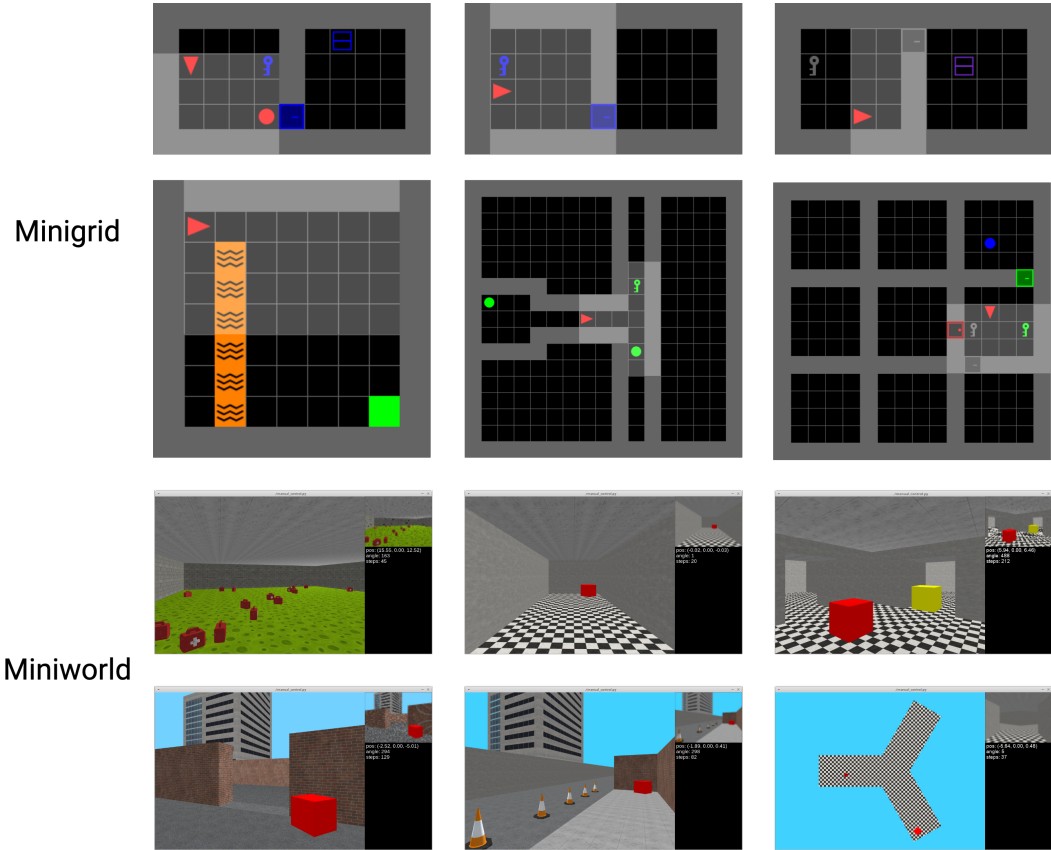

Figure 1: Example environments from Minigrid and Miniworld.

This paper outlines the Minigrid and Miniworld libraries for 2D and 3D goal-oriented environments which implement a suite of goal-oriented, navigation-based, and instruction-based environments. Furthermore, the two libraries have an easily extendable environment API for implementing novel research-specific environments. Example environments for both libraries are shown in Figure 1. In particular, Minigrid and Miniworld focus on providing users with the following features:

1. **Easy installation process** - The libraries maintain a minimal list of dependencies such that a wide range of audiences can easily use the libraries.

2. **Customizability** - Users can easily create new environments or add functionalities to existing environments.

3. **Easy to visualization** - The environments can be viewed from the top down, making it easier to visualize and understand the learned policy.

4. **Scalable complexity** - A range of environments with different complexity is provided, which allows users to understand the limitations of the learned policy.

The libraries can be installed using Python's package manager PIP (`pip install minigrid` and `pip install miniworld`) with environment documentation and tutorials available at `minigrid.farama.org` and `miniworld.farama.org`, respectively.

Minigrid and Miniworld have already been used for developing new RL algorithms in a number of areas, for example, safe RL [39], curiosity-driven exploration [24], and meta-learning [12]. Furthermore, research has built upon Minigrid for new environments, e.g. BabyAI [6] where the environment is constructed and evaluated using natural language-based instructions. However, despite the popularity of the libraries, to date, no academic paper has explained the design philosophy, environment API, or provided a case study for users.

# 2    Minigrid & Miniworld Libraries

In this section, we outline the design philosophy (Section 2.1), the environment specifications of Minigrid and Miniworld (Sections 2.2 and 2.3) along with the environment API (Section 2.4). Finally, we detail how published research has used both libraries to develop and evaluate novel Reinforcement Learning algorithms (Section 2.5).

The environments in the two libraries are partially-observable Markov Decision Processes (POMDP) [18]. These environments can be mathematically described by the tuple $(\mathcal{X}, \mathcal{A}, \mathcal{O}, \mathcal{T}, \mathcal{R}, \Omega, \gamma)$ where $\mathcal{X}$ represents the state space, $\mathcal{A}$ the action space, $\mathcal{O}$ the observation space, $\mathcal{T} : \mathcal{X} \times \mathcal{A} \to \mathcal{X}$ the transition function, $\mathcal{R} : \mathcal{X} \times \mathcal{A} \to \mathbb{R}$ the reward function, $\Omega : \mathcal{X} \to \mathcal{O}$ the observation function, and $\gamma \in [0, 1)$ the discount factor.

## 2.1    Design Philosophy

Minigrid and Miniworld were originally created at Mila - Québec AI Institute to be primarily used by graduate students. Due to the variety in usages, customizability and simplicity were the highest priority to allow as many users to use and understand the codebase. To support this, Python and Gym's RL environment API [5] (now updated to `Gymnasium` due to Gym no longer being maintained) was selected to implement the libraries due to their popularity within the machine learning and reinforcement learning communities. Example code for interacting with the environment is provided in Listing 1.

```python
import gymnasium as gym

# load the environment in upper-left corner of Figure 1
env = gym.make("MiniGrid-BlockedUnlockPickup-v0", render_mode="human")

observation, info = env.reset(seed=42)
for i in range(1000):
    # User-defined policy function
    action = policy(observation)
    observation, reward, terminated, truncated, info = env.step(action)

    if terminated or truncated:
        observation, info = env.reset()
env.close()
```

Listing 1: Code snippet for testing an RL policy in a Minigrid environment.

An additional core design point was to intentionally have as few external dependencies as possible, as fewer dependencies make these packages easier to install and less likely to break. As a result, Minigrid uses NumPy for the GridWorld backend along with the graphics to generate icons for each cell. Miniworld uses Pyglet for graphics with the environments being essentially 2.5D due to the use of a flat floorplan, which allows for a number of simplifications compared to a true 3D engine. This allows the libraries to run relatively fast but more importantly enables users to understand the whole environment implementations and customize them for their own needs.

## 2.2    Minigrid Environments

Each Minigrid environment is a 2D GridWorld made up of $n \times m$ tiles where each tile is either empty or occupied by an object, e.g., a wall, key, or goal. Using different tile configurations, tasks of varying complexity can be constructed. By default, the environments are deterministic with no randomness in the transition function ($\mathcal{T}$).

By default, agent observations ($\mathcal{O}$) are a dictionary with three items: "image", "direction", and "mission". Example environments with their corresponding "image" and "mission" are provided

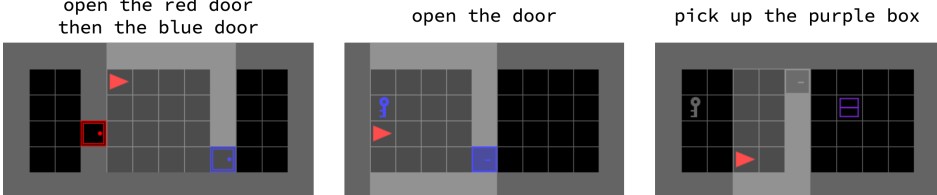

Figure 2: Example Minigrid environments with their mission instruction. For each of the environments, the highlighted region indicates the partial observation received by the agent.

in Figure 2. The "image" observation is a top-down render of the agent's view which can be limited to a fixed distance or of the whole environment. The direction is an integer representing the direction the agent is facing. The "mission" is a text-based instruction specifying the task to solve. The "mission" can change on each environment's reset as the goal might change, for example, "Minigrid-GoToObject" has a mission of "go to the {color} {obj_type}" where color can be one of ["red", "green", "blue", "purple", "yellow", "grey"] and obj_type can be one of ["key", "ball", "box"]. As a result, the instructions can be encoded as a one-hot vector. For more complex instructions, a language model is required to interpret for an RL agent.

The agents have a discrete action space ($\mathcal{A}$) of seven options representing ["turn left", "turn right", "move forward", "pickup", "drop", "toggle", "done"]. These actions are consistent across all environments however some actions might not produce any effect in certain states, e.g. "pickup" will not do anything if the agent is not next to an object that can be picked up.

The default reward function ($\mathcal{R}$) for the environment is sparse such that the reward is only non-zero when the mission is accomplished. Furthermore, the function can be easily customized for specific user needs through overwriting the environment's MiniGridEnv._reward function.

## 2.3 Miniworld Environments

Each Miniworld environment is a 3D world that consists of connected rooms with objects inside (e.g. box, ball, or key). Like Minigrid, the worlds can be configured for various tasks with different goals and complexity.

For the agent, the observation space ($\mathcal{O}$) is, by default, an RGB image of size $80 \times 60$ from the agent's perspective of the world. This image size can be modified by passing obs_width and obs_height arguments to the environment constructor. Five of the images in Figure 1 are example observations (with the final being a top-down view of the agent in an environment). To act in the world, agents are provided with a similar action space ($\mathcal{A}$) to Minigrid with an additional move-back action. Thus, there are in total eight discrete actions: ["turn left", "turn right", "move forward", "move back", "pickup", "drop", "toggle", "done"]. Like Minigrid, the default reward function is sparse with the agent only being rewarded when the agent completes the environment goal but can be modified in custom environments.

## 2.4 Constructing and Extending Environments

In both libraries, the environments can be created using a small set of functions. To demonstrate this feature, we showcase two sample scripts used to define the simulation environment in Listing 2, one for Minigrid and one for Miniworld.

The structure of the environment generation function is the same for more complex scenarios, with a few more helper functions. We have created tutorials for new environment creation: https://minigrid.farama.org/main/content/create_env_tutorial and https://miniworld.farama.org/main/content/create_env respectively. Both libraries can be directly integrated with existing RL libraries, e.g., Stable-Baselines3 (SB3). Additionally, to augment the libraries, we have created extra wrappers that customize the behavior of the libraries, such as adding stochastic actions and varying observation spaces, https://minigrid.farama.

```python
def _gen_grid(self, width, height):          def _gen_world(self):
    """Minigrid Example"""                       """Miniworld Example"""
    # Create an empty grid                        # Create a rectangular room
    self.grid = Grid(width, height)               self.add_rect_room(min_x=0,
    # Generate surrounding walls                  ↪   max_x=self.size, min_z=0,
    self.grid.wall_rect(0, 0, width,              ↪   max_z=self.size)
    ↪   height)                                   # Place goal in a random location
    # Place goal                                  self.box = self.place_entity(
    self.put_obj(Goal(), width - 2,                   Box(color="red")
    ↪   height - 2)                               )
    # Place agent in a random location            # Place agent in a random location
    self.place_agent()                            self.place_agent()
```

Listing 2: Code snippet for environment generation in Minigrid (left) and Miniworld (right).

`org/api/wrapper/` and `https://github.com/Farama-Foundation/Miniworld/blob/master/`
`miniworld/wrappers.py`.

## 2.5   Adoption

Since their creation, Minigrid and Miniworld have been widely adopted by the RL research community and used for various applications. Together, the two repositories have around 2400 stars and 620 forks on GitHub. We detail several instances where the two libraries have been utilized effectively.

**Curriculum Learning**: The two libraries provide a programmatic approach to creating new environments on-the-fly, this functionality can be utilized for automatic environment generation. For example, Dennis et al. [7] generated a natural curriculum of increasingly complex environments and Parker-Holder et al. [26] harnessed the power of evolution in a principled, regret-based curriculum.

**Exploration**: The reward function in the two libraries is, by default, a sparse reward making them ideal candidates for developing new exploration techniques. Using Minigrid and Miniworld, Seo et al. [31] developed an exploration approach using state entropy as the extrinsic reward and Zhang et al. [40] proposed a simple yet effective exploration criterion by equally weighting the novel areas.

**Meta Learning & Transfer Learning**: Given the ease of creating new simulation environments, the two libraries have also been used in developing meta-learning and transfer-learning algorithms. In Igl et al. [15], Minigrid has been used to develop regularization techniques to encourage agents to generalize to new environments. In Liu et al. [21], Miniworld is used to develop a new meta-learning approach that avoids local optima in end-to-end training, without sacrificing optimal exploration and Hutsebaut-Buysse et al. [14] explored the use of pre-trained task-independent word embedding for transfer learning.

Recent work has also used both libraries on a number of other research topics, demonstrating how Minigrid and Miniworld prove useful for a wide variety of domains. For example, Gumbsch et al. [9] leverage the partial observability in many of the Minigrid environments to develop POMDP planning algorithms; Zhou and Li [42] use Minigrid environments for inverse reinforcement learning tasks; and Zhao et al. [41] customized Minigrid to develop model-based RL algorithms.

## 3   Case Studies for Utilizing the Unified API

In this section, we provide two case studies to demonstrate the utility and ease of use of Minigrid and Miniworld's unified API and hope to inspire future studies that span both environments. The first is on RL agent transfer learning between different observation spaces of the two libraries. The second case study shows how human transfer learning can be conducted between different observation spaces.

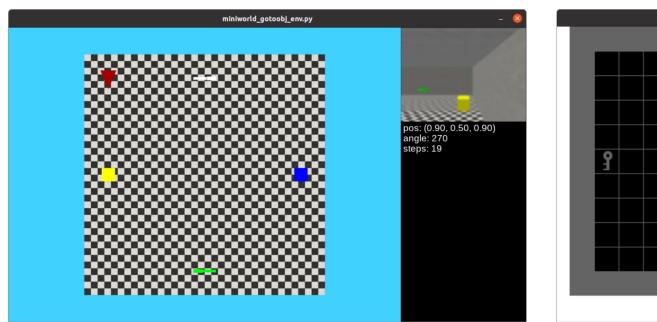

Figure 3: Visualization of the `miniworld-gotoobj-env` (left) and `minigrid-gotoobj-env` (right). The `miniworld-gotoobj-env` image shows both the top-down view and the agent view (top-right window). During training the agent only has access to the agent view of the environment.

## 3.1 RL Agent Transfer Learning Between Different Observations Spaces

In this case study, we showcase the ability to transfer policies learned on Minigrid to Miniworld. We created two similar simulation environments in Minigrid and in Miniworld, where the task is to follow an instruction to go to an object, we call the two environments `minigrid-gotoobj-env` and `miniworld-gotoobj-env` (screenshots of the two environments are shown in Figure 3). In both environments, the agent is given an instruction to "go to the {color} {object}" with the color and object being randomly selected from ["blue", "green", "grey", "purple", "red", "yellow"] and ["ball", "box", "key"], respectively.

When transferring the learned weights, one key question is which part of the agent's weights should be transferred. We first trained a PPO agent [30] on `minigrid-gotoobj-env` and then we transferred the learned weights to the PPO agent for `miniworld-gotoobj-env`. The PPO policy consists of a mission instruction encoder, an image encoder, an actor network, and a critic network. The policy transfer is made easy due to the unified APIs for Minigrid and Miniworld. We tested with 12 different weight transfer options, the results are given in Table 1.

To measure the effectiveness of the transfer learning, we define the transfer improvement as

$$\texttt{Transfer\_Improvement} = \frac{\texttt{Transfer\_Learning\_AUC} - \texttt{Miniworld\_Learning\_AUC}}{\texttt{Miniworld\_Learning\_AUC}} \quad (1)$$

where `Transfer_Learning_AUC` represents the area under the curve (AUC) of the reward curve for the agent that is initialized with Minigrid learned weights, and `Miniworld_Learning_AUC` represents the AUC of the reward curve for the agent with randomly initialized weights. Both the transfer learning agent and the Miniworld learning agent are trained for 200k time steps. As Table 1 shows: (1) the transfer learning behavior was improved when the critic network and mission embedding weights were not frozen; (2) transferring only the critic network and mission embeddings produces better results compared with also transferring the actor network weights.

## 3.2 Transfer Learning Between Different Observations Spaces for 10 Human Subjects

In this case study, we show how we can use the Minigrid and Miniworld libraries to collect and visualize human data. We utilized two similar environments in the Minigrid and Miniworld libraries, where there are four rooms and the goal is to reach a target position denoted with a green box, in the least amount of steps. In both cases, the human subject has only partial observation of the environment. We performed two sets of experiments. The first set of experiments lets the subject collect experience in the Minigrid environment for 10 episodes, then transfers to the Miniworld environment, and plays for another 10 episodes. The second set of experiments directly asks the subject to play on the Miniworld environment for 10 episodes. In both the Minigrid and Miniworld environments, the action space has dimension three with the actions: turn left, turn right, and go forward. To make the human experience more similar to the RL agent, we randomly assign the

| Transferred Weights | Non-Frozen Weights | | Frozen Weights | |
|---|---|---|---|---|
| | Mean (%) | STD (%) | Mean (%) | STD (%) |
| M | 0.089 | 6.760 | -3.775 | 8.421 |
| A | -8.881 | 14.399 | -4.670 | 14.997 |
| C | 3.993 | 3.189 | 2.760 | 2.703 |
| AM | -20.668 | 13.915 | -13.199 | 9.634 |
| CM | 3.207 | 2.808 | 0.001 | 3.957 |
| ACM | -9.494 | 9.217 | -30.958 | 16.540 |

Table 1: Transfer improvement for the 12 sets of experiments. For the transferred weights, "M": represents mission embedding weights, "A": represents actor network weights, and "C": represents critic network weights. "**Frozen Weights**" refers to freezing the transferred weights, while "Non-Frozen Weights" refers to not freezing the transferred weights.

three actions to the 1-9 numbers keys on the keyboard. The average rewards over 10 episodes on the Miniworld environments are shown in Table 2 and a sample subject trajectory is shown in Figure 4. Similar to what is shown in Figure 4, we empirically observe an adaption phase to the random key assignment during the first episode for every human subject.

| Subject No. | Minigrid ⇒ Miniworld | | Subject No. | Directly Miniworld | |
|---|---|---|---|---|---|
| | Mean | STD | | Mean | STD |
| 1 | 0.93 | 0.04 | 6 | 0.89 | 0.04 |
| 2 | 0.84 | 0.28 | 7 | 0.91 | 0.04 |
| 3 | 0.74 | 0.37 | 8 | 0.94 | 0.04 |
| 4 | 0.89 | 0.04 | 9 | 0.82 | 0.28 |
| 5 | 0.93 | 0.05 | 10 | 0.94 | 0.04 |

Table 2: Subject mean reward for the two sets of experiments. The "**Minigrid ⇒ Miniworld**" refers to the first set of experiments, and "**Directly Miniworld**" refers to the second set of experiments.

## 3.3 Implementation Details

In this section, we discuss how the case studies were implemented using the Minigrid and Miniworld libraries. The RL agent transfer learning case study represents a custom experimental setting not natively supported by Minigrid, Miniworld, and SB3. Thus, on top of the two libraries, we implemented the following functionalities:

1. created the `minigrid-gotoobj-env` (26 lines for `_gen_grid()`) and `miniworld-gotoobj-env` (19 lines for `_gen_world()`) environments;

2. augmented `miniworld-gotoobj-env` with mission instructions (3 lines of code similar to the `_gen_mission()` from Minigrid);

3. created a custom wrapper for `minigrid-gotoobj-env` to process the mission instructions (10 lines of code that are highly similar to the `ImgObsWrapper` in Minigrid);

4. created a custom feature extractor in SB3 for `minigrid-gotoobj-env` (23 lines mostly copied from the SB3 `NatureCNN` class);

5. created and trained a PPO agent on `minigrid-gotoobj-env` using SB3 (6 lines);

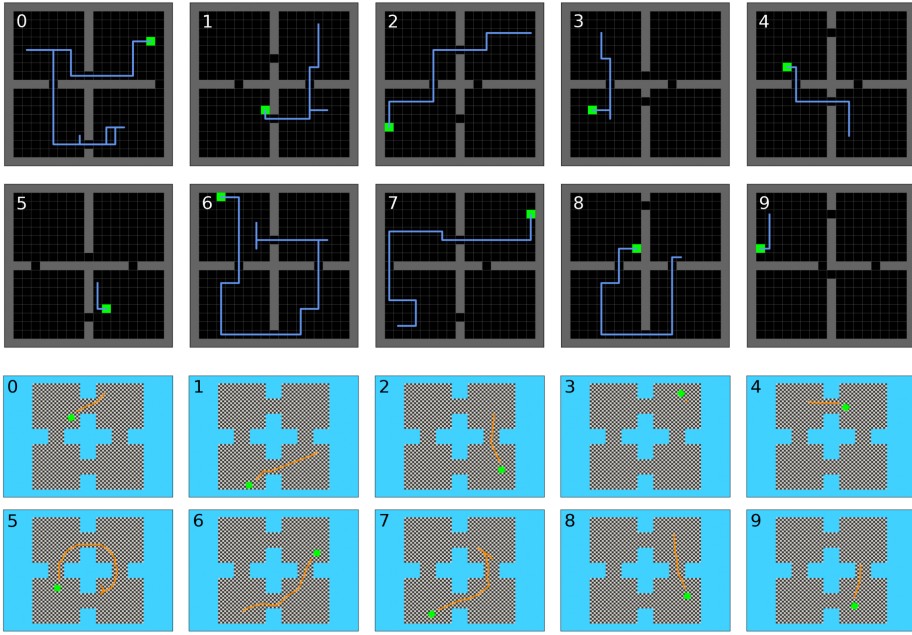

Figure 4: Trajectories from one human subject when testing transferring experience on Minigrid environments to Miniworld. The numbers correspond to the episode number.

6. transfer the learned policy from `minigrid-gotoobj-env` to `miniworld-gotoobj-env` (14-51 lines depending on the settings, can be reduced using a for loop).

In total, the implementation of this highly customized case study required 101 - 138 lines of code, highlighting the facility of use that our design provides. For the human transfer learning case study, we utilized two existing simulation environments, namely `MiniGrid-FourRooms-v0` and `MiniWorld-FourRooms-v0`. The human action input is achieved using the already existing manual control feature. To render the human trajectories, we altered the existing manual control feature and included data recording and data plotting functionalities. Although additional effort was required, given the Pythonic design of the two libraries, it only required three hours of coding. A detailed list of the required implementations is provided in the supplementary materials. For research with similar purposes, we plan to open-source our case study implementations as a separate codebase.

## 4   Related Works

Simulation libraries have been a crucial part of DRL research since the success of playing Atari games using deep RL [25]. However, most simulation environments focus on fixed, single tasks, such as swinging up a pendulum or making a humanoid stand up. Extending these environments to support custom objectives (e.g. "use the key to open the door and then get to the goal" is difficult).

In the robot learning community, there has been an increase in the number of benchmark simulation environments that focus on goal-oriented tasks, notably the pixmc environments [28] and the Franka kitchen environment [10]. Despite their popularity within the robot learning community, for RL research that solely focuses on the decision-making process, these robotic simulation benchmarks might not be the best experimental platform. One key issue is that these robot simulation benchmarks often utilize physics simulators, e.g., MuJoCo [37] and IssacSim [22], which are significantly more difficult to extend than Minigrid and Miniworld.

Historically, the RL community has made ample use of GridWorld-like environments for their research and education (notably by Sutton and Barto [35]). Given their wide usage, there have also been simulation libraries that focus on 2D GridWorld-like environments. MazeBase [34] is a simulation

library for GridWorld-like 2D games. However, because it is written in Lua and does not support the OpenAI gym API, it is difficult to integrate with existing deep learning and DRL libraries (e.g., PyTorch [27] and SB3 [29]). Griddly [2] is another library that provides GridWorld environments with a highly optimized and flexible game engine. Although Griddly provides more functionalities than Minigrid, it comes at the cost of higher complexity, making it difficult to understand and customize. DeepMind Lab2D [4] and Melting Pot [20] are also 2D GridWorld simulation environments. However, compared to Minigrid, they mainly focus on multi-agent reinforcement learning tasks and do not support human language instructions. Crafter [13] and some of the Minigrid environments (e.g., `MiniGrid-MultiRoom`) are both designed to evaluate the generalization, exploration, and long-term reasoning ability of RL agents. However, Crafter environments are more difficult compared to Minigrid environments. For RL researchers, Minigrid environments are better suited in the initial development phase of their algorithms, while Crafter environments can be used when the algorithm is more mature.

For Miniworld, the most relevant work is ViZDoom. The ViZDoom research platform [19] is a set of simulation environments based on the popular first-person shooter (FPS) game Doom that enables RL agents to make tactical and strategic decisions. At a high level, the type of environments that can be created are comparable in Miniworld and ViZDoom. However, ViZDoom uses a custom-designed language to create new scenarios, while Miniworld uses a small set of Python functions for environment creation. This makes Miniworld easier to use for the RL research community, which is more familiar with Python. Nevertheless, ViZDoom does support shooting games, depth information, and audio which are not supported by Miniworld. Another 3D simulation library with a similar purpose is DeepMind Lab [3], but given that the game engine is written in C and the levels are written using Lua, there is a steep learning curve for customizing it. 3D simulation environments like Habitat 3D [23, 36] and Unity-based simulation environments [17] also enable the agent to navigate and interact with its surroundings. Compared to Miniworld, they can simulate more complex dynamics and are more photo-realistic. However, for target-reaching and object-collection tasks where the goal is not deployment on embodied AI systems, Miniworld provides a much simpler and lightweight simulation solution that enables faster iteration and evaluation of new research ideas. Open-ended 3D environments like Avalon [1], MineRL [11], Malmo [16], and MineDojo [8] also have similar capabilities. But their focus is more on evaluating the RL agent's ability to generalize on a wide range of tasks, while Miniworld environments mostly focus on a single task.

# 5 Conclusion

The Minigrid and Miniworld libraries provide modular and customizable RL environments for goal-oriented tasks. We detailed the design philosophy behind the two libraries and provided a walkthrough of their API along with research areas that already utilize the two libraries. In our case studies, we have shown the unified API among two libraries provides an easy way to study transfer learning between different observation spaces and human decision-making. In future works, we plan to further develop the libraries' capabilities for human-in-the-loop decision-making.

**Limitations**: The libraries have two main limitations, first, the environment creation process prioritizes simplicity with minimal functions, which limits the type of environments that can be created. Second, the two libraries are implemented in Python, which makes them computationally slower than environments that utilize highly-optimized game engines in C++.

**Societal Impact**: Since the libraries have idealized system dynamics, the learned policy might not be directly applicable to real-world applications without introducing safeguard mechanisms.

# Acknowledgements

Minigrid and Miniworld were originally created as part of research work done at Mila - Québec AI Institute. We thank Manuel Goulão for their contribution to the documentation website.

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
