# Minigrid & Miniworld: Modular & Customizable Reinforcement Learning Environments for Goal-Oriented Tasks Supplementary Materials

**Maxime Chevalier-Boisvert**
Mila - Québec AI Institute
maximechevalierb@gmail.com

**Bolun Dai**
New York University
& Farama Foundation
bolundai@nyu.edu

**Mark Towers**
University of Southampton
& Farama Foundation
mt5g17@soton.ac.uk

**Rodrigo de Lazcano**
Farama Foundation
rperezvicente@farama.org

**Lucas Willems**
Miple
lucas.willems@miple.co

**Salem Lahlou**
Mila - Québec AI Institute
lahlosal@mila.quebec

**Suman Pal**
Telekinesis
suman7495@gmail.com

**Pablo Samuel Castro**
Google DeepMind
psc@google.com

**Jordan Terry**
Farama Foundation
& Swarm Labs
jkterry@umd.edu

## A    Dataset Documentation & URL

The source code of Minigrid and Miniworld can be found at `https://github.com/Farama-Foundation/Minigrid` and `https://github.com/Farama-Foundation/Miniworld` along with their documentation at `https://minigrid.farama.org/` and `https://miniworld.farama.org/`.

## B    Implementation Details for Transfer Learning Between Different Observations Spaces for 10 Human Subjects

To run the experiments, we have implemented the following functionalities:

1. implemented the human trajectory saving for `MiniGrid-FourRooms-v0` (copied the `ManualControl` class from Minigrid and added 38 lines of code, which are mostly calling data saving functions);
2. implemented the human trajectory saving for `MiniWorld-FourRooms-v0` (copied the `ManualControl` class from Miniworld and added 45 lines of code, which are mostly calling data saving functions);
3. implemented data saving and plotting for `MiniGrid-FourRooms-v0` (33 lines of code, mostly for Matplotlib);
4. implemented data saving and plotting for `MiniWorld-FourRooms-v0` (33 lines of code, mostly for Matplotlib).

In total, the implementation of this new functionality required 149 lines of code.

# C Hosting, Licensing, and Maintenance Plan

The source code is hosted on GitHub. Both the Minigrid and Miniworld libraries have Apache-2.0 licenses. The two libraries are planned to be maintained by the Farama Foundation in the foreseeable future, please refer to `https://farama.org/project_standards` for details.

# D Author Statement

We bear all the responsibility in case of violation of rights. Both libraries are under Apache-2.0 licenses.

# E Case Study Implementation

The implementation of the RL agent transfer learning case study can be found at `https://github.com/BolunDai0216/MinigridMiniworldTransfer`. The implementation of the Human transfer learning case study can be found at `https://github.com/BolunDai0216/MiniworldRecordData` and `https://github.com/BolunDai0216/MinigridRecordData`.

For our RL agent transfer learning case study, we used a single NVIDIA RTX A4000 GPU. When training the RL agent, we used the default Stable-Baselines 3 hyperparameters for the PPO algorithm. The default PPO hyperparameters can be found at `https://stable-baselines3.readthedocs.io/en/master/modules/ppo.html`.

# F Case Study Learning Curves

This section provides the learning curves used to compute the result shown in Table **??**.

# G Human Transfer Learning Experiment Instructions

| Experiment | Instruction |
|---|---|
| **Minigrid Pre-Train** (*Minigrid ⇒ Miniworld*) | *The purpose of this experiment is to enable the agent to reach a goal, you will realize what the goal is when you see it. You can control the agent's movement using the number keys 1-9, however, I do not know the functionality of each of the keys. The game will reset after you reach the goal. Please play this game for 10 rounds.* |
| **Miniworld Training** (*Minigrid ⇒ Miniworld*) | *Now we transfer to another set of environments. The control keys are the same as the previous experiments, the purpose is also the same. Please play this game also for 10 rounds.* |
| **Miniworld Training** (*Directly Miniworld*) | *The purpose of this experiment is to enable the agent to reach a goal, you will realize what the goal is when you see it. You can control the agent's movement using the number keys 1-9, however, I do not know the functionality of each of the keys. The game will reset after you reach the goal. Please play this game for 10 rounds.* |

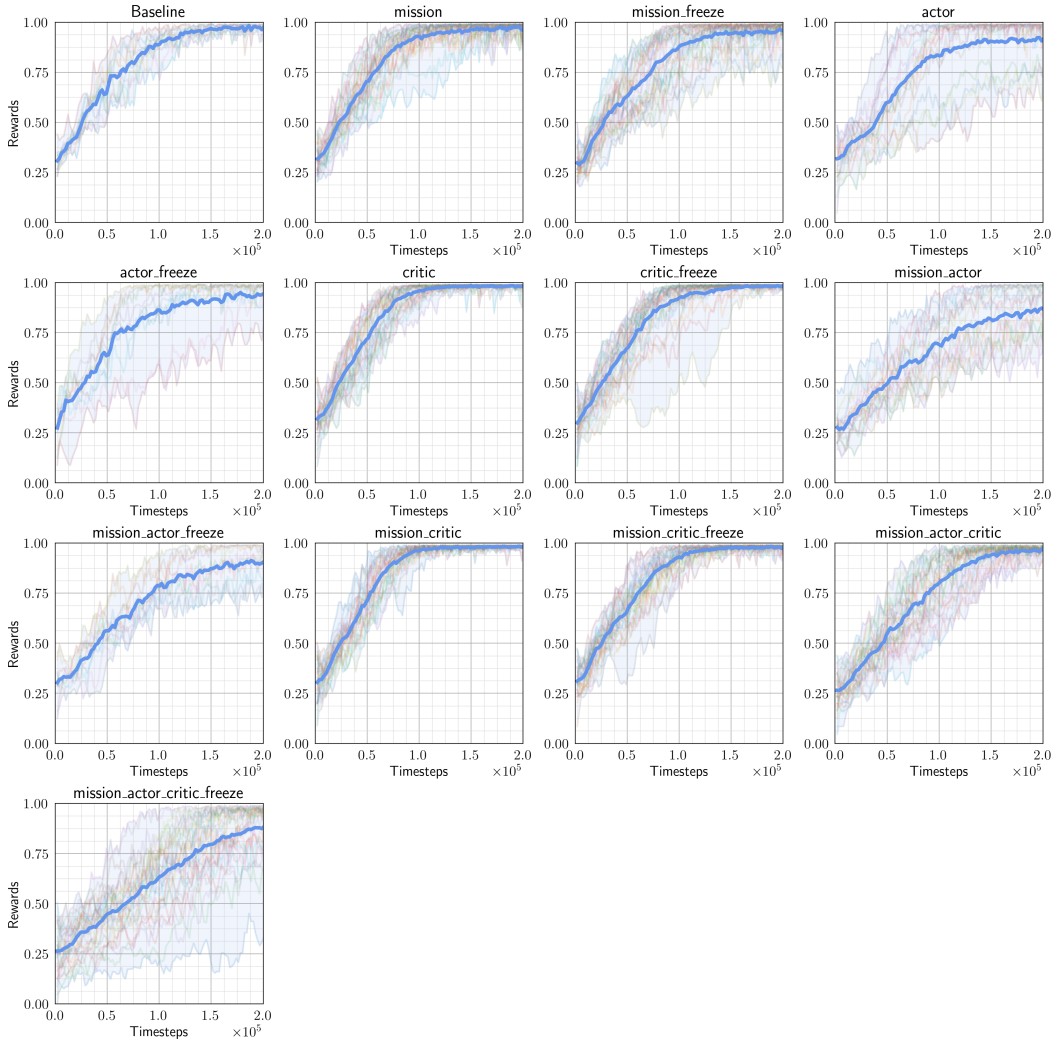

Figure 1: This figure shows the learning curves used to compute the transfer improvements. The solid blue curve is the mean reward curve over all the runs. The light blue region represents the region contained by the maximum and minimum reward at each timestep. The rewards for each individual run are given by the light-colored curves.

# H  Participant Compensation

The participants were all volunteers, thus, there was no monetary compensation. The total amount of money spent on participant compensation is $ 0.

# I  GitHub Stars & Citations

The Minigrid and Miniworld libraries have been widely used by the RL community. To date, the two libraries have around 2400 stars on GitHub and the number of stars is still increasing as shown in Figure 2. The two libraries have also been widely used as experimental platforms for RL research, this is evident by the 470 citations (based on Google Scholar) the Minigrid library has received. All of the aforementioned data are recorded on June 12, 2023.

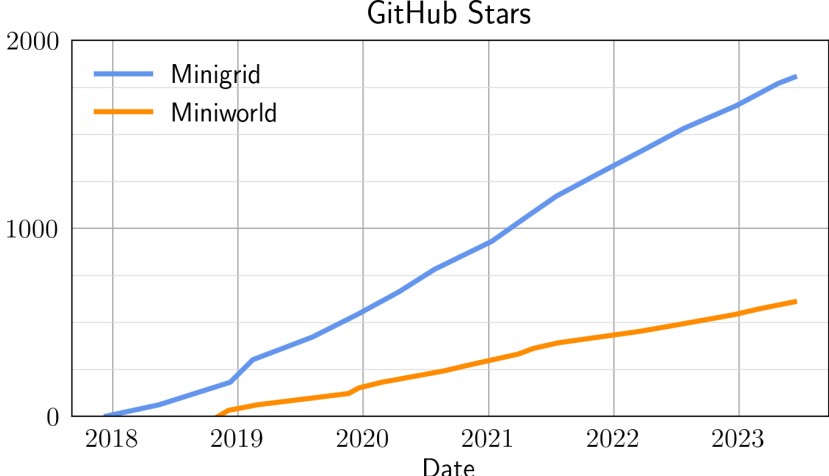

Figure 2: GitHub Stars evolution for Minigrid and Miniworld (recorded on June 12th, 2023, data obtained using `https://star-history.com`)

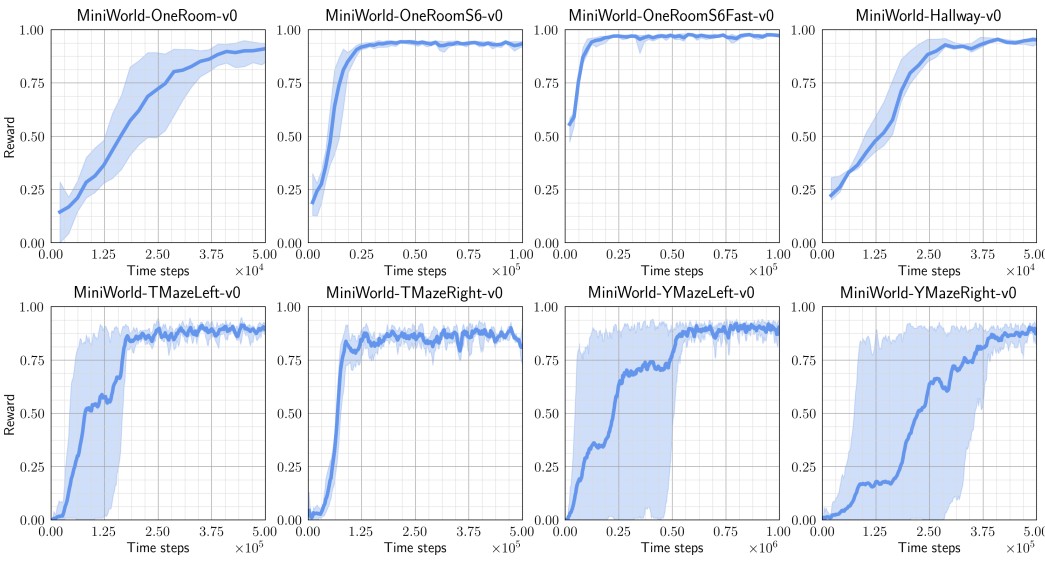

Figure 3: Baseline results for a set of Miniworld environments. The color scheme follows Fig. 4.

# J    Baseline Results

This section includes baseline results for a partial set of Minigrid and Miniworld environments, additional baseline results will be provided in the future on the documentation website[1]. The Baseline results are obtained after five runs of PPO [2] with the default hyperparameter settings of SB3 [1] and seeds [123, 124, 125, 126, 127].

---

[1]Minigrid: `minigrid.farama.org` and Miniworld: `miniworld.farama.org`

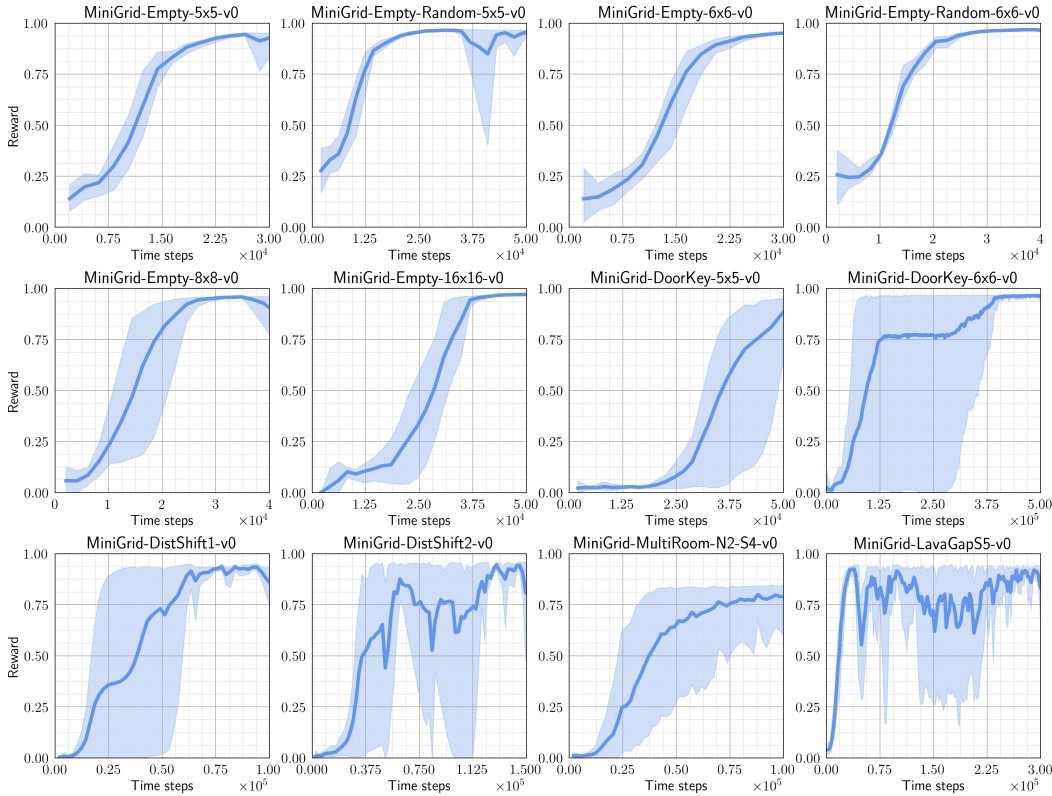

Figure 4: Baseline results for a set of Minigrid environments. The solid blue curve is the mean reward curve over all the runs. The light blue region represents the region contained by the maximum and minimum reward at each timestep.