# OpenReview forum: "Minigrid & Miniworld: Modular & Customizable Reinforcement Learning Environments for Goal-Oriented Tasks"
_NeurIPS.cc/2023/Track/Datasets_and_Benchmarks — NeurIPS 2023 Datasets and Benchmarks Poster_

### Official Review · Reviewer_Semn · 2023-07-17

**Rating:** 7
**Confidence:** 5
**Clarity:** Yes.

**Strengths:**

- The paper is well presented and easy to follow.
- Code is open-sourced. Documents are comprehensive.
- Two presented libraries are well-adopted by the community.
- The philosophy of minimalistic design makes the proposed libraries easy to extend, as supported by the case study.

**Additional Feedback:**

See above.

**Correctness:**

Baseline results on tasks in Minigrid and Miniworld are encouraged to provide such that following works are easy to compare against.

**Documentation:**

Good.

**Ethics:**

N/A.

**Limitations:**

See above.

**Opportunities For Improvement:**

- Authors are encouraged to further discuss existing works that adopt Minigrid and Miniworld.
- Authors are encouraged to discuss other related DRL environments/simulations. For example, DeepMind Lab 2D [Beattie et al., 2020] is a 2D version of the popular DeepMind Lab simulation. Melting Pot [Leibo et al., 2021] is another 2D environment built on top of DeepMind Lab 2D to study multi-agent RL. Open-ended environments are also worth discussing and comparing. These include but not limit to, for example, Malmo [Johnson et al., 2016] and MineRL [Guss et al., 2019] are simulations built on top of the video game Minecraft, which provides complex 3D worlds. Crafter [Hafner, 2022] is a simplified 2D version of Minecraft, testing a wide range of capabilities of learned agents. Avalon [Albrecht et al., 2022] provides procedurally generated 3D environments to test RL generalization. MineDojo [Fan et al., 2022] builds on top of MineRL and provides internet-scale data for policy learning in Minecraft.

References:
- Beattie et al., DeepMind Lab2D, 2020.
- Leibo et al., Scalable Evaluation of Multi-Agent Reinforcement Learning with Melting Pot, 2021.
- Johnson et al., The Malmo Platform for Artificial Intelligence Experimentation 2016.
- Guss et al., MineRL: A Large-Scale Dataset of Minecraft Demonstrations, 2019.
- Hafner, Benchmarking the Spectrum of Agent Capabilities, 2022.
- Albrecht et al., Avalon: A Benchmark for RL Generalization Using Procedurally Generated Worlds, 2022.
- Fan et al., MineDojo: Building Open-Ended Embodied Agents with Internet-Scale Knowledge, 2022.

**Relation To Prior Work:**

See above.

**Summary And Contributions:**

This paper presents Minigrid and Miniworld libraries, two simulated environments for reinforcement learning on goal-oriented tasks. This paper also provides case studies on the advantages of unified APIs between Minigrid and Miniworld. Source code is provided. Their usage is also well documented.

---

> ### Author Response · Authors · 2023-08-18
> **Response to reviewer Semn**
>
> We would like to thank the reviewer for their insightful comments and constructive criticism. We appreciate that the reviewer recognizes the benefits of the projects and their strengths.
>
> > Authors are encouraged to further discuss existing works that adopt Minigrid and Miniworld.
>
> In addition to the related works we have provided, there are many works that use a variety of Minigrid and Miniworld environments for different tasks. Minigrid has been used by Zhou et al., [ICML 2022] in developing algorithms for inverse reinforcement learning and by Gumbsch et al., [NeurIPS 2021] in developing POMDP planning algorithms. A custom variant of MiniGrid has been used by Zhao et al., [NeurIPS 2021] in developing model-based RL algorithms. These works along with many others demonstrate how Minigrid and Miniworld environments can be utilized and customized for different research topics. We will add this discussion to our submission.
>
> References:
> - Zhou et al., A Hierarchical Bayesian Approach to Inverse Reinforcement Learning with Symbolic Reward Machines, ICML, 2022
> - Zhao et al., A Consciousness-Inspired Planning Agent for Model-Based Reinforcement Learning, NeurIPS, 2021
> - Gumbsch et al., Sparsely Changing Latent States for Prediction and Planning in Partially Observable Domains, NeurIPS, 2021
>
> > Authors are encouraged to discuss other related DRL environments/simulations
>
> DeepMind Lab2D, Melting Pot, and Minigrid are similar in the sense that they all are 2D gridworld simulation environments. However, there are two key differences:
>
> - DeepMind Lab2D and Melting Pot focus on multi-agent reinforcement learning tasks, while Minigrid focuses on single-agent tasks.
> - Minigrid supports human language instructions.
>
> Crafter and some of the Minigrid environments (e.g., MiniGrid-MultiRoom) are both designed to evaluate the generalization, exploration, and long-term reasoning ability of RL agents. However, Crafter environments are more difficult compared to Minigrid environments. For RL researchers, Minigrid environments are better suited in the initial development phase of their algorithms, while Crafter environments can be used when the algorithm is more mature.
>
> Thus, we believe that Minigrid, Crafter, DeepMind Lab2D, and Melting Pot are complementary environment suites.
>
> Avalon, MineRL, Malmo, and MineDojo are all 3D simulation environments that evaluate the RL agent’s ability to generalize on a wide range of tasks. On the other hand, Miniworld environments mostly focus on a single task, e.g., target reaching and object collection. Thus, we believe that Miniworld, Avalon, MineRL, Malmo, and MineDojo are complementary to each other and can both be used by the RL research community.
>
> We will add this discussion to our submission.
>
> > Baseline results on tasks in Minigrid and Miniworld are encouraged to provide such that the following works are easy to compare against.
>
> Thank you for your suggestion. Since there are more than 100 registered environments in Minigrid and Miniworld, given our time and computation limitations, at this stage, we only include baseline results for a set of environments in the supplementary section of the updated manuscript. However, we do plan to provide baseline results for all Minigrid and Miniworld environments in the future.
>
> Baseline results for Minigrid environments can be obtained using the [`rl-starter-files`](https://github.com/lcswillems/rl-starter-files) and the code we provided [here](https://github.com/BolunDai0216/MinigridMiniworldTransfer/blob/main/test/minigrid_cnn_train_test.py) and, for environments that use the mission string, [here](https://github.com/BolunDai0216/MinigridMiniworldTransfer/blob/main/test/minigrid_door_test.py). Baseline results for Miniworld can be obtained using the code we provided [here](https://github.com/BolunDai0216/MinigridMiniworldTransfer/blob/main/test/miniworld_test.py).

---

> > ### Comment · Reviewer_Semn · 2023-08-21
> >
> > Thank you authors for the reply. I believe this submission becomes stronger after incorporating baseline results and further discussion with related works. Therefore I raised my score from 6 to 7.

---

### Official Review · Reviewer_FdB5 · 2023-07-21
**Review for RL Environments, Minigrid and Miniworld**

**Rating:** 6
**Confidence:** 3
**Clarity:** It is easy to read and overall very w…

**Strengths:**

1. It's a learning environment that's already been utilized by many papers and has earned a number of Github Stars. I think this is because the environment is easy to install, the coding can be simplified, and it is highly customizable.
2. This paper provides experimental evidence of the impact of each module through transfer learning experiments between different environments.

**Additional Feedback:**

I believe it will be a better study if you suggest a task and provide a baseline.
I increase the score 5 to 6  because it has the advantage of a customizable and simple env based on python.

**Correctness:**

The paper offers suggestions for tasks, but no mention of baseline setup and experimentation.

**Documentation:**

Well-organized documentation is available via github.

**Ethics:**

No ethical issues.

**Limitations:**

1. It's not clear what you're trying to show by sharing the results of a transfer learning experiment with human subjects, and I don't think table 1 and 2 are meaningfully comparable.
2. In this environment, it is necessary to show that customization will be useful in the future by providing suggestions for tasks or problems that do not exist or are rare.

**Opportunities For Improvement:**

1. I think it has garnered quite a few stars and references, and it would be nice to see more papers in the 2.5 Adaptation section to make it more widely available.
2. Section 3.1 only shows the experimental results, it would be nice to report the learning curve used to calculate the AUC.
3. For each of the related studies on Minigrid and Miniworld, it is necessary to investigate a wider range of environments and explain the differences in detail.

**Relation To Prior Work:**

For Miniworld, you only compare ViZDoom and DeepMind Lab in related work, but I would like to see a more detailed comparison of ViZDoom's customizability. In addition, it is recommended to compare with a wider range of 3D environments (Habitat 3D, Unity Machine Learning Environment, etc.).

Szot et al, Habitat 2.0: Training Home Assistants to Rearrange their Habitat, NeurIPS 2021
Savva et al. Habitat: A platform for embodied ai research. In: Proceedings of the IEEE/CVF international conference on computer vision. 2019.
JULIANI, Arthur, et al. Unity: A general platform for intelligent agents. arXiv preprint arXiv:1809.02627, 2018.

**Summary And Contributions:**

This paper proposed reinforcement learning environments based on design philosophy, which are goal-oriented 2D (Minigrid) and 3D (Miniworld), respectively. These environments are characterized by a simple installation process because they pursue few external dependencies as much as possible, and the learning environment is configured for easy customization and visualization. This paper presented an evaluation of the learning impact of each module of neural networks while performing transfer learning between minigrid and miniworld. In the experiment, the evaluation was done by the AUC of the learning curve, and this study reported that there was an improvement in transfer learning when the mission embedding and critic networks were non-frozen.

---

> ### Author Response · Authors · 2023-08-18
> **Response to reviewer FdB5 [Part 1]**
>
> We would like to thank the reviewer for their insightful comments and constructive criticism. We appreciate that the reviewer recognizes the benefits of the projects and their strengths.
>
> > I think it has garnered quite a few stars and references, and it would be nice to see more papers in the 2.5 Adaptation section to make it more widely available.
>
> In addition to the works that adopt Minigrid and Miniworld we have already provided, there are many works that use a variety of Minigrid and Miniworld environments for different tasks. Minigrid has been used by Zhou et al., [ICML 2022] in developing algorithms for inverse reinforcement learning and by Gumbsch et al., [NeurIPS 2021] in developing POMDP planning algorithms. A custom variant of MiniGrid has been used by Zhao et al., [NeurIPS 2021] in developing model-based RL algorithms. These works along with many others demonstrate how Minigrid and Miniworld environments can be utilized and customized for different research topics. We will add this discussion to our submission.
>
> **References**:
> - Zhou et al., A Hierarchical Bayesian Approach to Inverse Reinforcement Learning with Symbolic Reward Machines, ICML, 2022
> - Zhao et al., A Consciousness-Inspired Planning Agent for Model-Based Reinforcement Learning, NeurIPS, 2021
> - Gumbsch et al., Sparsely Changing Latent States for Prediction and Planning in Partially Observable Domains, NeurIPS, 2021
>
> > Section 3.1 only shows the experimental results, it would be nice to report the learning curve used to calculate the AUC.
>
> We have included the learning curves in the supplementary materials of the updated manuscript.
>
> > It's not clear what you're trying to show by sharing the results of a transfer learning experiment with human subjects, and I don't think table 1 and 2 are meaningfully comparable.
>
> The intent of the transfer experiments was not to directly compare the performance of RL transfer learning and human transfer learning. What we intended to demonstrate with these experiments is the ease with which they can be performed between Minigrid and Miniworld environments due to the unified API. Our hope was that this example can inspire future studies that span both environments, by leveraging the unified API. We will clarify this point in the text.
>
> > For Miniworld, you only compare ViZDoom and DeepMind Lab in related work, but I would like to see a more detailed comparison of ViZDoom's customizability.
>
> The customizability of Miniworld and ViZDoom are similar. Both of the environments are essentially 2.5D (more than 2D but not really 3D environments). Therefore, at a high level, the type of environments that can be created are comparable. One of the main advantages that Miniworld has against ViZDoom is that the customization process is much simpler for RL researchers. ViZDoom uses ACS scripting, which is a language specifically created for the Doom engine and is similar to C in syntax. While on the other hand, Miniworld uses only a small set of Python functions for environment creation. This makes Miniworld significantly easier to use by the RL research community, which is more familiar with Python. Nevertheless, we do acknowledge that, with the price of a steeper learning curve, ViZDoom does support shooting games, depth information, and audio which are not supported by Miniworld. Therefore, ViZDoom and Miniworld can be seen as _complementary_ to each other. We will add this discussion to our submission.

---

> > ### Author Response · Authors · 2023-08-18
> > **Response to reviewer FdB5 [Part 2]**
> >
> > > For each of the related studies on Minigrid and Miniworld, it is necessary to investigate a wider range of environments and explain the differences in detail. In addition, it is recommended to compare with a wider range of 3D environments (Habitat 3D, Unity Machine Learning Environment, etc.).
> >
> > DeepMind Lab2D, Melting Pot, and Minigrid are similar in the sense that they all are 2D gridworld simulation environments. However, there are two key differences:
> >
> > - DeepMind Lab2D and Melting Pot focus on multi-agent reinforcement learning tasks, while Minigrid focuses on single-agent tasks.
> > - Minigrid supports human language instructions.
> >
> > Crafter and some of the Minigrid environments (e.g., MiniGrid-MultiRoom) are both designed to evaluate the generalization, exploration, and long-term reasoning ability of RL agents. However, Crafter environments are more difficult compared to Minigrid environments. For RL researchers, Minigrid environments are better suited in the initial development phase of their algorithms, while Crafter environments can be used when the algorithm is more mature.
> >
> > Miniworld, Habitat 3D, and Unity-based simulation environments all focus on 3D simulation environments where the agent is able to navigate and interact with its surroundings, there are a few key differences:
> > - Habitat 3D and Unity-based environments are able to simulate rigid-body dynamics via the use of physics engines, while Miniworld only allows certain types of movements (forward, backward, turning) and interactions with its environment (for example, picking up and dropping objects).
> > - Habitat 3D and Unity-based environments are full 3D simulation environments, while Miniworld is essentially 2.5D.
> > - Habitat 3D and Unity-based environments are more photo-realistic compared to Miniworld.
> >
> > On the other hand, for target reaching and object collection tasks where the goal is not deployment on embodied AI systems, Miniworld provides a much simpler and lightweight simulation solution that enables faster iteration and evaluation of new research ideas. Thus, we see Habitat 3D and Unity-based simulation environments as complementary environment suites to Miniworld.
> >
> > We will add this discussion to our submission.
> >
> > > In this environment, it is necessary to show that customization will be useful in the future by providing suggestions for tasks or problems that do not exist or are rare. I believe it will be a better study if you suggest a task and provide a baseline.
> >
> > The usefulness of Minigrid and Miniworld's customizability is already evident in many existing works where custom variants of Minigrid and Miniworld environments are used to study a wide range of research problems. For example, Zhao et al., [NeurIPS 2021] created custom variants of Minigrid environments in developing model-based RL algorithms; Ding et al., [NeurIPS 2022] create a custom variant of `MiniGrid-Unlock` to study how variational causal reasoning can be used be generalize goal-conditioned RL; Liu et al., [ICML 2021] created a new environment in Miniworld for studying meta-reinforcement learning, this environment (`MiniWorld-Sign`) is now included in the Miniworld library. Klissarov et al., [NeurIPS 2020] use a custom Miniworld environment called `MiniWorld-MyWayHome-v0` to study reward shaping using graph convolutional networks. In addition to these previous works, the case studies we have provided also suggest a new task for Minigrid and Miniworld.
> >
> > **References**:
> > - Zhao et al., A Consciousness-Inspired Planning Agent for Model-Based Reinforcement Learning, NeurIPS, 2021
> > - Ding et al., Generalizing Goal-Conditioned Reinforcement Learning with Variational Causal Reasoning, NeurIPS, 2022
> > - Liu et al., Decoupling Exploration and Exploitation for Meta-Reinforcement Learning without Sacrifices, ICML, 2021
> > - Klissarov et al., Reward propagation using graph convolutional networks, NeurIPS, 2020

---

> > > ### Comment · Reviewer_FdB5 · 2023-08-21
> > > **Response**
> > >
> > > This paper addresses many of my concerns, including the various citations and the learning curve.
> > >
> > > Based on the responses so far, the environment proposed in the paper is expected to be slower than ViZDoom but easily customizable, faster than Habitat 3D or Unity-based simulations, but much simpler.
> > > It will also lack a physics engine and interaction with objects compared to MuJoCo.
> > > The paper doesn't compare rendering speeds, so it may not be accurate.
> > >
> > > I believe that this paper would be much better with a more forward-looking problem definition and proposal, and with environments or characteristics not previously covered.
> > >
> > > While there are definitely some shortcomings, I would raise the score to a 5 or 6 because it has the advantage of a customizable env based on python, at an early stage for experimentation, and because of this usability advantage.

---

### Official Review · Reviewer_2jWC · 2023-07-22
**Review of Miningrid&Miniworld**

**Rating:** 4
**Confidence:** 4
**Clarity:** The paper is well written and there a…

**Strengths:**

The main strength of the tool lies in its simplicity. It adopts very few libraries. The user can train the tasks without any deep knowledge about RL. The ability to create new environments from the code is also interesting. I appreciate the formalisation of the relation among reward observation, state space and action space. The tool is very helpful for beginners in this area as everything runs out of the box.

**Additional Feedback:**

I do not think that the tool is suitable as a benchmark tool in RL community. I really appreciate the effort to simplify the process of RL training for newcomers. On the other hand there is not any possibility how to use the tool for scientific purposes. I do not see any way how to extend the paper to fit the needs of scientific benchmark tool for the community.

**Correctness:**

The benchmarks are correct. I lack some table summarising the accuracy of trained networks.

**Documentation:**

There is documentation presented

**Ethics:**

There are not any ethical issues. There are some data collected from the humans but without any ethical concerns.

**Limitations:**

The authors state clearly the limitation of the tool. It is more educational tool  than scientific benchmark.

**Opportunities For Improvement:**

The tool is suitable for educational purposes but it lacks the features for research in the RL. It can help to understand basic principles of RL but the user will soon face the limits of this tool. Although it is possible to customise the environments and tasks, the lack of physics, object interaction and capability of agent are the main drawbacks. I do not see any possibility how to adopt this tool for research purposes. It is not possible to adopt trained networks to a real world tasks (authors mention this limitation in the text). There is also slow library for RL training adopted (Stable Baselines3) so the training of more complex tasks will be problematic.

**Relation To Prior Work:**

The authors stems from the previous work. The tool was developed for more many years.

**Summary And Contributions:**

The paper presents a tool to train RL agents in simple environments consisting either from grid or simplified world. The tool is focused on ability to create and train agents from few lines of code and visualise and understand the process of training and evaluation. There are several tasks in each environment presented that can be customised and extended programatically. The authors provides simple baselines for  transfer learning based on transfer improvement metrics. There is also data from 10 subjects present while data was collected by teleoperation.

---

> ### Author Response · Authors · 2023-08-18
> **Response to reviewer 2jWC**
>
> We would like to thank the reviewer for their insightful comments and constructive criticism. We appreciate that the reviewer recognizes the benefits of simplicity and the environment's formalism of the projects.
>
> > The tool is suitable for educational purposes but it lacks the features for research in the RL. It can help to understand basic principles of RL but the user will soon face the limits of this tool. Although it is possible to customize the environments and tasks, the lack of physics, object interaction and capability of agents are the main drawbacks. I do not see any possibility of adopting this tool for research purposes. It is not possible to adopt trained networks to real world tasks (authors mention this limitation in the text). I do not think that the tool is suitable as a benchmark tool in the RL community. I really appreciate the effort to simplify the process of RL training for newcomers. On the other hand there is not any possibility how to use the tool for scientific purposes. I do not see any way to extend the paper to fit the needs of scientific benchmark tools for the community.
>
> While the simplicity of Minigrid and Miniworld is a key feature of the two libraries, we would strongly object that this makes it incapable of being used by RL researchers. In evidence for this, we note that a prior software citation of Minigrid (Chevalier-Boisvert et al.) alone has collected over 500 citations showing that a wide number of papers have cited and used the projects within academic work already. Second, this is backed up by Section 2.5 where we list a number of peer-reviewed papers that actively use Minigrid and Miniworld for testing. So in addition to helping newcomers, the simplicity of the environments allows RL researchers to segment and identify problems when developing and testing reinforcement learning algorithms.
>
> Furthermore, the customizability of the environments makes it quite appealing and relevant for testing a number of capabilities of autonomous agents. For example, work has been done using Minigrid and Miniworld for developing new algorithms in inverse reinforcement learning [Zhou et al., ICML 2022], curriculum learning [Dennis et al., NeurIPS 2020; Parker-Holder et al., ICML 2022], long-term planning [Xu et al., NeurIPS 2019], model-based reinforcement learning [Zhao et al., NeurIPS 2021], planning in POMDPs [Gumbsch et al., NeurIPS 2021].
>
> Furthermore, the prior software-only citation for Minigrid alone has accumulated over 500 citations.
>
> Of course, the use of Minigrid does not preclude the use of other benchmarks for evaluation; however, targeted evaluations of agent capabilities can be more difficult in non-modifiable environments like the ones in many standard benchmarks. Thus, we believe Minigrid and Miniworld are complementary to other benchmarks and strengthen the evaluation of new research.
>
> References:
> - Chevalier-Boisvert et al., *Minimalistic gridworld environments for OpenAI Gym*, Github repository, 2018
> - Zhou et al., *A Hierarchical Bayesian Approach to Inverse Reinforcement Learning with Symbolic Reward Machines*, ICML, 2022
> - Dennis et al., *Emergent complexity and zero-shot transfer via unsupervised environment design*, NeurIPS, 2020
> - Parker-Holder et al., *Evolving curricula with regret-based environment design*, ICML, 2022
> - Xu et al., *Regressing Planning Networks*, NeurIPS, 2019
> - Zhao et al., *A Consciousness-Inspired Planning Agent for Model-Based Reinforcement Learning*, NeurIPS, 2021
> - Gumbsch et al., *Sparsely Changing Latent States for Prediction and Planning in Partially Observable Domains*, NeurIPS, 2021
>
> > There is also slow library for RL training adopted (Stable Baselines3) so the training of more complex tasks will be problematic.
>
> Could the reviewer please clarify what they mean by a “slow library for RL training”? Although perhaps not optimized for wall-time efficiency, stable baselines3 has become a common library for RL research with more than 1.3k forks and more than 6.3k stars on GitHub, and 723 citations (based on Google Scholars as of Aug 12th, 2023).
>
> > It lacks some table summarizing the accuracy of trained networks.
>
> Could the reviewer please clarify what they mean by “accuracy of the trained networks”? Since RL is not optimizing towards a stationary target (as in supervised learning), there is no clear notion of “accuracy”. Thus, performance curves are typically used as proxies for evaluating the quality of what the agent has learned. We will add in the supplementary materials of the updated manuscript the training performances for a set of Minigrid environments using a baseline PPO agent.

---

> > ### Author Response · Authors · 2023-08-23
> > **Response to reviewer 2jWC**
> >
> > Thank you once again for your careful review. Although we feel we have addressed your concerns, we want to confirm with you whether there are additional issues you feel need to be addressed. Given that only a few days are left for discussion with the authors, we want to ensure we have enough time to properly respond to your concerns.

---

> > ### Comment · Reviewer_2jWC · 2023-08-28
> > **Reply to the authors**
> >
> > Dear authors,
> >
> > thanks for the clarification.
> >
> > "Of course, the use of Minigrid does not preclude the use of other benchmarks for evaluation; however, targeted evaluations of agent capabilities can be more difficult in non-modifiable environments like the ones in many standard benchmarks. Thus, we believe Minigrid and Miniworld are complementary to other benchmarks and strengthen the evaluation of new research."
> >
> > I do agree that your benchmark tool is suitable for testing novel RL algorithms, however recent advances in RL allows to train more complex tasks and sim2real transfer. The tasks presented in your toolbox does not guarantee its scalability towards more complex tasks.
> >
> > "Furthermore, the prior software-only citation for Minigrid alone has accumulated over 500 citations."
> >
> > Can you please explain what is advantage of your new benchmark over old Minigrid publication? Why it requires separate publication?
> >
> > "Could the reviewer please clarify what they mean by a “slow library for RL training”? Although perhaps not optimized for wall-time efficiency, stable baselines3 has become a common library for RL research with more than 1.3k forks and more than 6.3k stars on GitHub, and 723 citations (based on Google Scholars as of Aug 12th, 2023)."
> >
> > Most of the recent RL benchmark tools allows to parallelize both simulated environments and policy updates. Recent trends in fast policy implementations allows to train billions of steps in days and improve the success rate (e.g. Rllib for training and IsaacSim or Galactic for environment).
> >
> >
> > "Could the reviewer please clarify what they mean by “accuracy of the trained networks”? Since RL is not optimizing towards a stationary target (as in supervised learning), there is no clear notion of “accuracy”. Thus, performance curves are typically used as proxies for evaluating the quality of what the agent has learned. We will add in the supplementary materials of the updated manuscript the training performances for a set of Minigrid environments using a baseline PPO agent."
> >
> > Sorry for the typo. I meant the success rate. I would like to see the performace curves for the Miniworld tasks.

---

> > > ### Author Response · Authors · 2023-08-29
> > > **Response to reviewer 2jWC**
> > >
> > > > I do agree that your benchmark tool is suitable for testing novel RL algorithms, however recent advances in RL allows it to train more complex tasks and sim2real transfer. The tasks presented in your toolbox does not guarantee its scalability towards more complex tasks.
> > >
> > > >Most of the recent RL benchmark tools allow parallelization of both simulated environments and policy updates. Recent trends in fast policy implementations allow us to train billions of steps in days and improve the success rate (e.g. Rllib for training and IsaacSim or Galactic for environment).
> > >
> > > Although it is true there exists more complex environments, they are more computationally expensive. Thus, while these more complex environments can be useful for showcasing scalability of new methods, they are unwieldy when performing initial explorations and deeper analyses, ablations, etc. Our libraries allow the user to control the dimensionality of the problems while providing useful tools for exploration and analysis of RL methods. Again, this is all complementary to the more complex methods you mention. Additionally, Minigrid and Miniworld environments can be ran in parallel using the Vector API in [Gymnasium](https://gymnasium.farama.org/api/vector/).
> > >
> > > > Can you please explain what is the advantage of your new benchmark over the old Minigrid publication? Why does it require separate publication?
> > >
> > > The citation numbers we were referring to are for the GitHub repo. The submitted manuscript includes previous unmentioned details, i.e., explanation of the design philosophy, explanation of the environment API, and providing case studies for users. Additionally, since its original release, Minigrid has undergone a significant refactor, in particular resulting in a unified API with Miniworld. As we demonstrate in the paper, this unified API grants researchers greater flexibility in exploring goal-directed domains of varying sizes, observation spaces, and difficulty, in addition to transfer across environment modalities.
> > >
> > > > Sorry for the typo. I meant the success rate. I would like to see the performance curves for the Miniworld tasks.
> > >
> > > We have included reward curves for a set of Miniworld environments in the supplementary material section of the updated manuscript.

---

### Official Review · Reviewer_UmzS · 2023-07-24
**Modular & Customizable Reinforcement Learning Environments for Goal-Oriented Tasks**

**Rating:** 7
**Confidence:** 4
**Correctness:** Yes
**Clarity:** The paper is well-written.

**Strengths:**

The proposed libraries offer several key advantages:

- Their minimalistic design paradigm makes these libraries user-friendly, streamlining both installation and customization.
- They accommodate a broad range of research topics, including but not limited to, Curriculum Learning, Exploration, Meta Learning, and Transfer Learning.
- The unified API between Minigrid and Miniworld enables transfer learning (for both RL agents and humans) between the different observation spaces.
- These libraries are supported by comprehensive documentation and well-maintained open-source code.


**Additional Feedback:**

N/A

**Documentation:**

Yes

**Limitations:**

The authors have discussed the limitations of this work in their paper. See the "Opportunities For Improvement" for my suggestions.

**Opportunities For Improvement:**

- Section 3.1 introduces the results of transfer learning between different observation spaces. It employs a metric known as "transfer improvement" to quantify the benefits of transfer learning. However, this scalar number alone doesn't fully illustrate the process. Including learning curves within the paper could provide a more comprehensive understanding.

**Relation To Prior Work:**

Yes

**Summary And Contributions:**

This paper, titled "Minigrid & Miniworld: Modular & Customizable Reinforcement Learning Environments for Goal-Oriented Tasks", presents the Minigrid and Miniworld libraries. These libraries provide a suite of goal-oriented 2D and 3D environments for reinforcement learning. They were explicitly created with a minimalistic design paradigm to allow users to rapidly develop new environments for a wide range of research-specific needs, leading to their widespread adoption.

---

> ### Author Response · Authors · 2023-08-18
> **Response to reviewer UmzS**
>
> We would like to thank the reviewer for their insightful comments and constructive criticism. We appreciate that the reviewer recognizes the key advantages of the projects.
>
> > Section 3.1 introduces the results of transfer learning between different observation spaces. It employs a metric known as "transfer improvement" to quantify the benefits of transfer learning. However, this scalar number alone doesn't fully illustrate the process. Including learning curves within the paper could provide a more comprehensive understanding.
>
> Thank you for your suggestion. As noted in the meta-review we will include the learning curves in the supplementary materials of the updated manuscript.

---

> > ### Author Response · Authors · 2023-08-23
> > **Response to reviewer UmzS**
> >
> > Thank you once again for your careful review. Although we feel we have addressed your concerns, we want to confirm with you whether there are additional issues you feel need to be addressed. Given that only a few days are left for discussion with the authors, we want to ensure we have enough time to properly respond to your concerns.

---

> > > ### Comment · Reviewer_UmzS · 2023-08-31
> > >
> > > Thank the authors for the response. I don't have further concerns at this point.

---

### Official Review · Reviewer_RPEh · 2023-07-24
**One of the best RL benchmarks**

**Rating:** 8
**Confidence:** 4
**Clarity:** I don't have any major concerns.

**Strengths:**

* Easy to use and update/reuse environments based on your research needs.
* MiniGrid's simplicity and interpretability make it a great playground for benchmarking RL algorithms and understanding their behavior.
* MiniWorld's realistic and visually immersive environments provide a more challenging and dynamic space for training agents in complex, real-world scenarios.
* Minimalistic design approach is easy to understand and start using for everyone.

**Additional Feedback:**

I think there are two main directions of the improvements:
1) work on complexity
2) work on performance, for example move envs to the gpu.

**Correctness:**

* Only one environment (Four Rooms was used). It would be nice to have more results from different environments together with training charts.
* Overall paper is well written and easy to read.

**Documentation:**

It was enough information to reproduce transfer learning examples.
All needed files could be found on github. I don't see any major issues.

**Ethics:**

I don't see any ethical concerns.

**Limitations:**

I don't see any negative social impact. Only limitations are related to the basic nature of the environments: simplified world simulation.

**Opportunities For Improvement:**

* It would be nice to rewrite using end2end gpu pipeline using something like warp from NVIDIA or jax from google. In this case it should be possible to have a few thousand of the environments running in parallel. Especially in case of the Miniworld performance is the largest concern.
* More information about tasks could be provided. For example maximum possible reward or how hard are them.

**Relation To Prior Work:**

Yes

**Summary And Contributions:**

Both MiniGrid and Miniworld are well known environments among a lot of ML researchers.
Number of stars in both repositories shows

* MiniGrid is a popular grid-world RL environment designed to evaluate the capabilities of various RL agents in a simple yet challenging setting. What stands out most about MiniGrid is its simplicity and easy-to-understand structure. The environment consists of grid-like rooms, each containing objects, agents, and goals. The agent's task is to navigate through the grid, interact with objects, and achieve specific objectives while avoiding obstacles.

* MiniWorld takes a more visually realistic approach to RL environments. MiniWorld provides 3D environments that simulate various real-world scenarios, offering a more immersive experience for agents. This environment diversity, ranging from indoor scenes to outdoor landscapes, greatly enriches the learning process. MiniWorld's realistic setting allows researchers to explore and address complex challenges present in real-world scenarios, like navigation in cluttered environments or handling dynamic objects. This is particularly beneficial when training RL agents for applications like robotic control or autonomous vehicles, where the agent needs to interact effectively with the real world.

* Transfer learning evaluation between so different domains sounds really interesting research direction.

---

> ### Author Response · Authors · 2023-08-18
> **Response to reviewer RPEh**
>
> We would like to thank the reviewer for their insightful comments and constructive criticism. We appreciate that the reviewer recognises the popularity and significant strengths of the projects.
>
> > Only limitations are related to the basic nature of the environments: simplified world simulation. **Additional Feedback**: Work on Complexity.
>
> We agree that the simplicity of Minigrid and Miniworld is one of their limitations but it is also one of their strengths and key features. In particular, the ability for users to test and develop new environments quickly and easily; an ability significantly more difficult for more complex simulation environments, i.e., MuJoCo. Furthermore, we are interested in expanding the projects to support more complex features for environments while preserving our strengths of simplicity and ease of use.
>
>
> > It would be nice to rewrite using an end2end gpu pipeline using something like warp from NVIDIA or jax from google. In this case it should be possible to have a few thousand of the environments running in parallel. Especially in the case of the Miniworld, performance is the largest concern. **Additional Feedback** Work on Performance, for example move environments to GPU.
>
> Using an End2End GPU pipeline would definitely improve the performance of the two libraries, but at the cost of library complexity, which makes it more difficult for users to customize and create environments. Furthermore, using an End2End GPU pipeline would also make the installation procedure more difficult, especially for students and newcomers to the reinforcement learning community. An alternative option for users wishing to increase sample speed is to use the Vector API in [Gymnasium](https://gymnasium.farama.org/api/vector/) to run Minigrid and Miniworld environments in parallel.
>
> > More information about tasks could be provided. For example maximum possible reward or how hard they are.
>
> The maximal possible reward for most environments can be obtained via a grid search algorithm such as A-star. However, due to the sparseness of the rewards and the use of mission strings, the difficulty of the environments varies significantly.  In the updated manuscript, we will provide baselines for each environment using a baseline PPO implementation which can serve as a proxy of the difficulty of each environment for users.
>
> > Only one environment (Four Rooms was used). It would be nice to have more results from different environments together with training charts.
>
> The aim of the case study is to show that transfer learning can be easily done between Minigrid and Miniworld environments due to the unified API. To recreate the Human Transfer experiment for other environments, the procedure will be exactly the same.
>
> In addition to the works that adopt Minigrid and Miniworld we have already provided, there are many works that use a variety of Minigrid and Miniworld environments for different tasks. `MiniGrid-DoorKey` has been used by Gumbsch et al., [NeurIPS 2021] in developing POMDP planning algorithms. `MiniGrid-KeyCorridor` has been used by Zhou et al., [ICML 2022] in developing algorithms for inverse reinforcement learning. A custom variant of `MiniGrid-CrossingEnv` has been used by Zhao et al., [NeurIPS 2021] in developing model-based RL algorithms. These works along with many others demonstrate how Minigrid and Miniworld environments can be utilized and customized for different research topics.
>
> **References**:
> - Zhou et al., *A Hierarchical Bayesian Approach to Inverse Reinforcement Learning with Symbolic Reward Machines*, ICML, 2022
> - Zhao et al., *A Consciousness-Inspired Planning Agent for Model-Based Reinforcement Learning*, NeurIPS, 2021
> - Gumbsch et al., *Sparsely Changing Latent States for Prediction and Planning in Partially Observable Domains*, NeurIPS, 2021

---

> > ### Author Response · Authors · 2023-08-23
> > **Response to reviewer RPEh**
> >
> > Thank you once again for your careful review. Although we feel we have addressed your concerns, we want to confirm with you whether there are additional issues you feel need to be addressed. Given that only a few days are left for discussion with the authors, we want to ensure we have enough time to properly respond to your concerns.

---

### Author Response · Authors · 2023-08-18
**Author Meta Response**

We thank all the reviewers for the valuable feedback. Based on the comments, we have enhanced the quality of the manuscript and helped clarify our ideas. We provide a detailed point-to-point response to all reviewers’ questions. We wish to point out the following in relation to common feedback we received.

The reviewers have pointed out the following strengths of the paper:

- The minimalistic design paradigm of Minigrid and Miniworld streamlines both the installation and customization process.
- The two libraries can accommodate a broad range of research topics, including but not limited to, Curriculum Learning, Exploration, Meta Learning, and Transfer Learning.
- The unified API between Minigrid and Miniworld enables transfer learning (for both RL agents and humans) between different observation spaces.
- The two libraries are supported by comprehensive documentation and well-maintained open-source code.

The reviewers had the following common concerns, which we would like to point out the following:

- **Further discussion regarding adoption**: The two libraries have already been widely adopted by the RL research community, which is evident by 500+ citations of prior software citations for Minigrid and over 3000 GitHub stars for the two projects. In addition to what has already been included in the manuscript, there are many works that use a variety of Minigrid and Miniworld environments for different tasks. For example, `MiniGrid-KeyCorridor` has been used by Zhou et al., [ICML 2022] in developing algorithms for inverse reinforcement learning; `MiniGrid-DoorKey` has been used by Gumbsch et al., [NeurIPS 2021] in developing POMDP planning algorithms; a custom variant of `MiniGrid-CrossingEnv`` has been used by Zhao et al., [NeurIPS 2021] in developing model-based RL algorithms.
- **Further discussion regarding related works**: 2D simulation libraries such as DeepMind Lab2D, Melting Pot, and Crafter are visually similar to Minigrid, however, they focus on different research problems and tasks. 3D simulation libraries such as Habitat 3D, Unity-based simulation environments, Avalon, MineRL, Malmo, and MineDojo also mainly focus on a different set of research problems. Of course, the use of Minigrid and Miniworld does not preclude the use of other benchmarks for evaluation; however, targeted evaluations of agent capabilities can be more difficult in the environments listed above. Thus, we believe Minigrid and Miniworld are *complementary* to other benchmarks and strengthen the evaluation of new research.
- **Learning curve for case study**: we will include the learning curves in the supplementary materials of the updated manuscript.

**References**

- Zhou et al., A Hierarchical Bayesian Approach to Inverse Reinforcement Learning with Symbolic Reward Machines, ICML, 2022
- Zhao et al., A Consciousness-Inspired Planning Agent for Model-Based Reinforcement Learning, NeurIPS, 2021
- Gumbsch et al., Sparsely Changing Latent States for Prediction and Planning in Partially Observable Domains, NeurIPS, 2021

---

### Decision · Program_Chairs · 2023-09-22

**Decision:**

Accept (Poster)

**Comment:**

MiniGrid & MiniWorld are environments released in 2018 that have been influential in RL research, with GitHubs having garnered 500+ citations in that time.  This paper is the much-belated first submission of these works to a conference, this time with their new maintainers, and with a unified API for the two environments and some experiments demonstrating domain transfer between the two.

**Perceived Strengths:**
* The simplicity and interpretability of these environments were highlighted by reviewers, especially for their use in benchmarking & designing new algorithms.
* Similarly ease of installation, comprehensive documentation and customizability of the environments were highlighted as positives by several reviewers.
* The extensive record of impact that these environments have had speaks for itself, especially in the field of curriculum design and others.

**Perceived Weaknesses:**
* One reviewer suggested this environment lacks features or complexity for research. Had this been an entirely new environment this might have been true, however other reviewerts argued that the simplicity was (and still is) a valuable feature of these environments.
* One reviewer questioned what improvements have been made to warrant a separate publication, while another suggested that they would prefer it if the work was more 'forward looking' highlighting new challenges. Authors have responded that this work has a unified API, that might be useful for transfer learning, and that this paper contains unmentioned details from the Github.

**AC View:**

This is an unusual paper to AC.  The core of this work in this submission was completed 5 years ago, and has already had a substantial impact on the field.  However, since it has never been officially published it is now presented here, with some modest adjustments, for acceptance in 2023. In these reviews we see that 'old' core of the work here is strongly praised by reviewers, while the response to 'new' experiments on transfer and unified API is somewhat cooler and more mixed.  While I am curious as to what motivated the submission of this widely-known work half a decade on, I must agree with the majority of reviewers in recognising the impact this environment has had on the field in advancing AI, even today.